# Effects of PCE on the Dispersion of Cement Particles and Initial Hydration

**DOI:** 10.3390/ma14123195

**Published:** 2021-06-10

**Authors:** Weiwei Zhu, Qingge Feng, Qi Luo, Xiukui Bai, Xianhao Lin, Zhao Zhang

**Affiliations:** 1School of Chemistry and Chemical Engineering, Guangxi University, Nanning 530004, China; zhuww1230@163.com; 2School of Materials and Environmental Engineering, Guangxi University for Nationalities, Nanning 530004, China; 3School of Resources, Environment and Materials, Guangxi University, Nanning 530004, China; ghost20050901@163.com (X.B.); guet_lin@163.com (X.L.); z1141466379@163.com (Z.Z.); 4School of Civil Engineering, Chongqing Jiaotong University, Chongqing 400074, China; qiluo@cqjtu.edu.cn

**Keywords:** PCE, water-to-cement ratios, dispersing properties, initial hydration

## Abstract

The effects of polycarboxylate superplasticizers (PCEs) on the dispersing properties and initial hydration of cement particles with various water-to-cement (w/c) ratios was investigated, including the water film thickness (WFT), rheology, fluidity, adsorption of PCEs, zeta potential, degree of hydration, hydration products. The experimental results demonstrate that the initial rheological and fluidity parameters were more sensitive to the PCE dosage at a lower w/c because the WFT and the zeta potential on cement particles change more significantly. Moreover, the higher adsorption amounts of the PCEs at a lower w/c lead to a stronger inhibition of the initial hydration, whilst, at the same PCE dosage, the cement pastes have a more rapid fluidity loss and quicker hydration reactions at a higher w/c due to a lower adsorption amount of the PCE on cement particles.

## 1. Introduction

The rheology properties of fresh cement paste are mainly dependent on the interactions between cement particles, the particle packing and the formation of early hydration products [1,2,3]. The use of superplasticizer (SP) can improve fluidity by virtue of its dispersion ability [4,5]. Comb-like polycarboxylate (PCE) superplasticizers have recently been developed and have become an integral part of modern concrete technology [6,7]. Their effects on the workability properties of fresh cement pastes (fcps), such as on the evolution of the microstructure and hydration process during the early hydration stage, still remain to be investigated [8,9,10,11]. Most studies focused on the relationship between the structure of PCEs and the fluidity of cement paste, the effect of the changing of WFT in the presence of PCEs on the fluidity of cement paste, as well as the influence of PCE on the growth rate of hydration products and the fluidity. However, there has been no systematic study about the adsorption behavior, interaction forces, the hydration products of nucleation and growth, and free water volume. In particular, reports on the effects of PCEs on the early hydration products are inconsistent.

Extensive studies have been conducted to clarify the key parameters controlling the performance of fcps. Additionally, water film thickness (WFT), defined as the ratio of the excess water to the total surface area, has been identified as one of the key factors [12]. After Kwan [13] established a new wet packing method for the determination of packing density, many studies have proved that the WFT is the main factor controlling the fresh properties of paste [14,15], mortar [16] and concrete [17]. Li et al. [18] found that the superplasticizer dosage and WFT are the key parameters determining the workability properties of fcps. Despite mounting efforts being carried out to investigate this topic, the detailed mechanism by which superplasticizers affect the fluidity of fcps remains unsolved. PCEs can release the water inside the flocculation and increase the free water content, thus improving fluidity of a cement paste significantly. At different w/c ratios, the changes in the adsorption amounts of PCEs and zeta potentials of fcps remain to be investigated. On the other hand, according to many reports, the voids between the large particles in the fcps significantly decrease in the presence of PCEs [19,20]. The hydration degree per unit volume of cement particles also changes due to significant changes in water volume during hydration. The effects of the water volume, zeta potential and hydration products on the fluidity that are realized by the PCEs must be identified and investigated.

On the other hand, extensive studies have been conducted to study the influence of the incorporation of PCEs into cement hydration processes. Puertas et al. [21] reported that the increase in PCEs dosages in fcps delayed the appearance of two hydration peaks, indicating that the addition of PCEs could retard initial cement hydration. Furthermore, the rheological results show that a small amount of PCE can lead to a huge reduction in the yield stress of the fcps. The presence of PCEs leads to changes in the amount of initial hydration products, especially the content of ettringite (AFt). Roncero et al. [22] and Jansen et al. [23] found in the study of the influence of PCEs on the hydration of the fcps that PCEs take effect within 15 min, at which time AFt could be detected in the fcps. Some researchers insisted that PCEs could significantly inhibit the formation of AFt crystals at the beginning of hydration [24,25]. To date, the effects of PCEs on hydration kinetics and hydration products during the induction period have not been well documented.

In addition, the effects of w/c on cement hydration have been well recognized. Hu et al. [26] investigated the influence of w/c on the heat of hydration, and they found that a lower w/c resulted in a higher heat of hydration rate at earlier hours but reduced rate after that, and the total heats of hydration within the first 24 h were almost the same among w/c values. Ley–Hernandez [27] and Kirby [28] provided some novel modified form of the phase boundary nucleation and growth models to explain why cement hydration rates remain insensitive to changes a cement paste’s water content. Can the cement hydration rates remain insensitive to changes in the paste’s water content in the presence of PCEs? This factor plays the most prominent role in determining their fluidizing effect, the hydration products, the zeta potential and the WFT.

This paper was launched to investigate the effects of the PCE at different w/c ratios on the dispersion properties and initial hydration of a cement paste. By measuring WFT, rheology, fluidity, adsorption amount, zeta potential, calorimetry, chemically bound water, X-ray diffraction (XRD) and thermogravimetric (TG) at different w/c ratios, the correlations of the dispersion capability of the PCE with zeta potential, WFT and the hydration products were ascertained. The results can be used to deeply analyze the combined effects of water content and PCE content.

## 2. Materials and Methods

### 2.1. Materials

Cement corresponding to Chinese standard GB8076-2008 type P·I 42.5 cement was used. Additionally, its composition is listed in Table 1. The fineness of the cement was 0.4%, and the density was 3.15 g·cm^−3^.

The cement was homogenized using a homogenizer and dried at 70 °C for three hours. A laser diffraction particle size analyzer was used to measure the grading curve of the cement and the results are presented in Figure 1.

A commercial polycarboxylate ether-based comb polymer that contained 50% solid matter was provided by Jiangsu Subote Chem. Reagent Company, Nanjing and was used as SP in the present study. The number average molecular weight (Mn), mass average molecular weight (Mw) and polydispersity index (Mw/Mn) were 20670 g/mol, 59733 g/mol and 2.9, respectively. In the following experiment, we diluted the 50% solid matter to 20%. The diluted solution was used in all experiments.

### 2.2. Mix Design

Firstly, 24 fresh cement pastes with the cement at selected PCE dosages and w/c ratios were prepared. The PCE dosage (by mass of cement) was varied in steps of 0.04% from 0.0% to 0.20% by weight of cement (bwoc) and for w/c ratios of 0.29, 0.35, 0.40 and 0.50. When mixing, water that contained various concentrations of PCEs and cement were added simultaneously, and the water that was contained in the PCEs was included.

For easy identification, each cement paste was assigned a serial code in the format of C–Y–Z, where C denotes the cement pastes, Y denotes the PCE dosage and Z denotes the w/c ratio. The codes are tabulated in the first column of Table 2.

### 2.3. Testing Methods

#### 2.3.1. Determination of the Packing Density and WFT

A wet packing test was established by Kwan et al. [29,30]. The WFT in a cement paste was calculated via the following steps: Step (1): the maximum solid concentration (ϕ_max_) of fcps was found by measuring six to eight samples at different w/c ratios plus the prescribed dosage of PCEs as the packing density of the cement. Step (2): the minimum void ratio (u_min_) was calculated as u_min_ = (1 − ϕ_max_)/ϕ_max_. Step (3): the water ratio (u_w_) was defined as the ratio of the volume of water to the volume of the cement. Step (4): the excess water ratio was calculated as (u_w_’) = u_w_−u_min_. Step (5): the specific surface area of the cement (A_c_) was confirmed by measuring the particle size distribution of the cement. Step (6): the WFT, which has the physical meaning of being the average thickness of the water films coating on the cement particles, was calculated as WFT = u_w_’/A_c_.

#### 2.3.2. Yield Stress and Plastic Viscosity

Rheological parameters were determined with concentric cylindrical geometry using a Paar Physica MCR 301 rheometer (Anton Paar, Graz, Steiermark, Austria). The measurement was performed 15 min after mixing by preshearing the paste at a shear rate of 100 s^−1^ for 1 min. A decreasing shear rate was then directly applied from 100 s^−1^ to 0 s^−1^ for 10 s. After being kept static for 1 min, shear stress was detected recording a flow curve with shear rates increasing from 0 up to 100 s^−1^ for 1 min and decreasing from 100 down to 0 s^−1^ for 1 min. As shown in Figure 2, the data points were fitted using the improved Bingham model. The rheological parameters were obtained from the following equation:(1)$$\tau =\tau_{0}+\mu \times \gamma +c\times \gamma^{2}$$
in which  τ, τ_0_, γ, μ, c are, respectively, the shear stress, yield stress, the shear rate, plastic viscosity, and a second-order parameter.

#### 2.3.3. Fluidity of Fresh Cement Pastes

The initial fluidity and fluidity evolution over time of fcps with adding PCEs were measured by a mini-cone test that complied with Chinese standard GB/T 8077. After mixing well, the cement paste was transferred into the truncated cone and then the cone was vertically removed quickly. After the paste stopped flowing, the diameter of spread cement paste was recorded. For the measurement of the fluidity evolution over time of the paste, wet rags were used to prevent the water content of the paste from evaporating, and it was stored for 30, 60, 90, and 120 min. Before testing the fluidity evolution over a predetermined time, to ensure homogeneity, the paste was stirred at a speed of 125 rpm for 60 s.

#### 2.3.4. Adsorption Amount

The adsorption isotherms of PCEs on the surfaces of mineral particles in fcps with w/c = 0.29 and w/c = 0.50 were evaluated by a total organic carbon (TOC) analyzer. At selected hydration times, the cement paste was immediately centrifuged at 8500 rpm for 10 min. Then, a clear supernatant solution was collected by using a syringe filter with a pore diameter of 0.22 μm. Subsequently, the supernatant solution was diluted with 0.01 M HCl and determined by the TOC analyzer (Shimadzu, Tokyo, Japan).

#### 2.3.5. Zeta Potential

Zeta potential of cement particles at different w/c ratios and P/Cs was measured using a Malvern Zetasizer Nano ZS90 (Malvern instruments, Malvern, England). The cement paste was diluted to a concentration of 0.3 wt.% by a saturated Ca(OH)_2_ solution. The calcium concentration of the saturated Ca(OH)_2_ solution was 20 mM and the pH value was 12. The concentration of PCE in the cement paste was equal to that of the TOC measurement.

#### 2.3.6. Isothermal Calorimeter

We introduced a novel method using self-design calorimetric equipment to investigate the first peak of cement hydration. This equipment was described in our patent [31]. A sketch of our calorimetric equipment is presented in Figure 3. The hydration occurred in a calorimetric cell, which was a stainless-steel cylinder. During the return to thermal equilibrium, 8.00 g of the cement powder was stored in a stainless-steel cup. Then, water that contained various concentrations of the PCE at the two w/c ratios was pushed down into the cup by a nozzle spring, and hydration began.

#### 2.3.7. Hydration Termination of the Hydrating Cement Pastes

Before the XRD and TG measurement of the hydrating cement pastes at predetermined hydration times, the hydration of the cement had to be stopped. At a predetermined hydration time, the hydrating sample was mixed with cold isopropanol (5 °C) using a solvent/solid ratio of 25. After stirring for 1 min, the liquid part of the cement–isopropanol suspension was removed by centrifugal force. The solid part after centrifugation was stored afterwards in a desiccator until constant mass. For thermogravimetric analysis, the obtained solid was washed again with diethylether, and the remaining isopropanol was removed.

#### 2.3.8. Chemically Bound Water

The chemically bound water was estimated by the difference in mass between 105 °C and 950 °C. The hydration times of these samples were 5 min, 30 min, 60 min, 90 min, and 120 min. The degree of hydration α is the ratio of the measured chemically bound water, namely EL, to that corresponding to 100% hydration, namely ${\mathrm{EL}}_{\infty }$.
(2)$$\alpha =\frac{\mathrm{EL}}{{\mathrm{EL}}_{\infty }}$$

Referring to the study of Powers and Brownyard [32], ${\mathrm{EL}}_{\infty }$ is considered to be equal to 25% per weight of anhydrous cement when OPC is used. For each cement, α is a mean value of three measurements of EL.

#### 2.3.9. XRD

Hydrated phase development of fcps at different w/c ratios and different PCE dosages at curing ages of 5 min and 120 min hydration was determined by XRD at room temperature using a Broker D8 Advance diffractometer (Bruker Axs Gmbh, Karlsruhe, Baden-württemberg, Germany) with monochromatic K_α_Cu radiation. The hydration termination method was the same as that used in the chemically bound water analysis. In this apparatus, the K_α_Cu radiation is generated in a Cu tube at 40 mA and 40 kV. The scan range was set from 5° to 75° (2θ, using a scanning speed of 2°/min.

#### 2.3.10. TG Analysis

TG analysis was used to examine early hydration’s products of the fcps that contained PCEs by a NETZSCH STA 449 F3 (NETZSCH, Saarbrücken, Germany). The hydration times of these samples were 5 min and 120 min. The sample (ca. 40 mg) was heated from 30 °C to 1000 °C with a heating rate of 10 °C/min under a nitrogen atmosphere.

## 3. Results and Discussion

### 3.1. Packing Density

The packing density results of fcps when adding different PCE dosages and different w/c ratios are tabulated in Table 2 and plotted in Figure 4. According to Figure 4, as the PCE dosage increased the packing density of the cement paste gradually increased. Specially, increasing the PCE dosage from 0.0% to 0.20% led to increase in the packing density of the fcp from 0.546 to 0.590. Thus, the packing density of the cementitious materials can be improved by the dispersion effect of the PCE.

### 3.2. WFT Results

The WFT of each paste was obtained by calculating the results of corresponding packing density, specific surface area of the cement particles, water ratio and excess water ratio. From the above, it is obvious that the addition of PCEs significantly increased the packing density, so in turn governs the excess water ratio. The excess water ratios of the fcps are listed in the fourth column of Table 2. A higher PCE dosage led to a larger water ratio and a larger excess water ratio. In addition, various w/c ratios also have a significant effect on water ratio. Additionally, a higher w/c ratio also led to a have water ratio and a larger excess water ratio. Additionally, the water ratio of the fcps is tabulated in the third column of Table 2. Since cement was the unique cementitious material in this research, and the specific surface areas of all samples were the unique and equal to 1,156,000 m^2^/m^3^.

Figure 5 displays the effects of the PCE dosages and the w/c ratios on the WFT. It is seen that the WFT increased as both the PCE dosage and the w/c ratio increased. This result is a combination of the corresponding changes in the packing density and water ratio. The results revealed that the WFT increased as the PCE dosages increased from 0.04% to 0.20%. In addition, for the w/c = 0.29 fcps, the WFT increased from 0.073 to 0.191 μm with the PCE dosage, while for the w/c = 0.50 fcps the WFT increased from 0.647 to 0.765 μm with the PCE dosage. As the PCE dosage increased from 0% to 0.20%, the WFT in w/c = 0.29 fcps increased up to 161.6%, but the WFT in w/c = 0.50 fcps only increased up to 18.2%. Therefore, an obvious increase in WFT occurs in the case of the lower w/c with increasing PCE dosage, whereas a slight increase is observed in the higher w/c case.

### 3.3. Rheology

The results of the yield stress and apparent viscosity are plotted against the PCE dosages for different w/c ratios in Figure 6. It can be seen that apparent viscosity decreased as the PCE dosages increased, thereby drastically reducing yield stress. In addition, the effects of the PCE on the yield stress and apparent viscosity of the fcps were dependent on the w/c ratio. As the w/c ratio increased, the marked difference in the yield stress and apparent viscosity of the fcps prominently decreased. For the w/c = 0.29 fcps, the yield stress decreased from 12.33 to 0.377 Pa and the apparent viscosity decreased from 1.877 to 0.606 Pas with the PCE dosage, while, for the w/c = 0.50 fcps, the yield stress decreased from 2.763 to 0.051 Pa and the apparent viscosity decreased from 0.469 to 0.028 Pas with the PCE dosage. The same trend is observed as indicated in the WFT variation with the w/c ratio. Additionally, the relationship between the rheological parameters and WFT is discussed in Section 3.5.1.

### 3.4. Flow Spread

The effects of the PCE dosages on flow spread with different w/c ratios are plotted in Figure 7. At w/c = 0.29, without the PCE, the paste cannot flow. In presence of the PCE, the paste can flow and increasing the PCE dosage from 0.004% to 0.2% increased the fluidity of fcps from 77 mm to 317 mm. On the other hand, at w/c = 0.50, without the PCE, the paste can flow and increasing the PCE dosage from 0.00% to 0.08% increased the fluidity of fcps from 198 mm to 340 mm. The initial fluidity at the lower w/c was more sensitive to the PCE dosage than that at the higher w/c, which is in accordance with the finding by Shui et al. that the fluidity was more sensitive to the PCE dosage at lower w/c than at higher w/c [33]. Furthermore, the above trend of the fluidity is similar in general to the trend of the WFT. This indicates that the incorporation of the PCE can effectively increase the WFT and improve the fluidity of the paste, because the WFT can lubricate the cement interface and reduces the friction between the cement particles [34,35]. Additionally, the relationship between flow parameters and the WFT is discussed in Section 3.5.2.

### 3.5. Water Film Thickness Effects

#### 3.5.1. Yield Stress and Apparent Viscosity versus WFT

The effects of WFT on the yield stress and apparent viscosity with various w/c ratios are plotted in Figure 8 and Table 3. It shows that the yield stress and apparent viscosity decreased as the WFT at a gradually increased rate. Additionally, it is obvious that the WFT has certain direct effect on the yield stress and apparent viscosity. The regression analysis yields a very high R^2^ above 0.9, indicating that an important factor affecting the rheological parameters of the cement paste is the WFT.

Furthermore, the variation of yield stress and apparent viscosity decreased with the WFT increased. Additionally, the WFT was highly dependent on the w/c ratio and the PCE dosage. At a higher w/c, the effect of the PCE dosage on the WFT was lower than that at a lower w/c, which is in accordance with the results presented by Kwan [36].

#### 3.5.2. Flow Spread versus WFT

The relationship between the flow spread and the WFT is shown in Figure 9 and Table 4. The R^2^ were well above 0.9 between the WFT and flow spread, indicating that the WFT was the important parameter affecting the flow spread, which is consistent with previous reported results [21]. Moreover, the variation of flow spread decreased with the WFT increased. As the WFT was highly dependent on the w/c ratio and the PCE dosage, the influence of the WFT on the flow spread is larger at a relatively low WFT than at a relatively high WFT. In addition, at a given WFT, the flow spread became significantly higher with a higher PCE dosage at a lower w/c ratio. Thus, the effect of PCE dosage on the fluidity of cement paste is also direct and positive, which is in accordance with the results presented by Kwan [36].

### 3.6. Impacts of PCE on the Fluidity Loss

The effects of various total PCE dosages on the fluidity loss of fcps with various w/c ratios are presented in Figure 10. As shown in Figure 10a, at w/c = 0.29, the fluidity of the fcps is successfully retained for 2 h by adding the PCEs at a dosage that exceeds 0.08% bwoc. With a lower dosage, the fluidity of the fcps slowly decreases within 2 h. As seen from Figure 10b, at w/c = 0.50, the fluidity of the fcps is successfully retained for 2 h by adding PCEs at any dosage. However, the fluidity of the fcps decreases substantially within 2 h with a lower dosage of PCEs, namely 0.04% bwoc. In these fcps with the same dosage of the PCEs, the fluidity of w/c = 0.50 fcps exhibits a much better dispersing effect than that of w/c = 0.29 fcps. However, the influences of the water amount and the hydration inhibition by the PCEs on fluidity of fcps have not been identified.

Another important desired property of PCEs is the fluidity retention of cement paste, since the workability of cement mortar and concrete must be maintained for a period of time. The fluidity retention of the PCEs in the two w/c pastes was examined. As shown in Figure 11, under the condition of the same PCE dosage, compared with the cement pastes with 0.50 w/c ratio, the pastes with 0.29 w/c ratio have smaller fluidity loss within 2 h. It should be noted that, in cement pastes with 0.50 w/c ratio, bleeding will occur when the dosage of PCE exceeds 0.08% bwoc. The fluidity loss of fcps is contributed primarily to the formation of early hydration products of the cement [37], namely the AFt crystals act as crosslinkers that bridge the cement particles, thereby leading to flocculation of the cement particles. Therefore, more AFt crystals in the 0.50 w/c system led to quicker fluidity loss.

The dispersion function of superplasticizers is thought to work only after their adsorption on the surfaces of cement particles. To identify the dispersing capability of PCEs at different w/c ratios, it is useful to compare their adsorption behaviors.

### 3.7. Adsorption Behavior of the PCEs and the Dispersing Power per Adsorption Amount on Cement

The PCEs can substantially influence their adsorption mode at various w/c ratios. The dispersion ability of the PCEs is attributed to the adsorption of PCE on the surfaces of cement particles, which generates electrostatic repulsion and steric hindrance between cement particles [38]. Thus, it is important to study the adsorption behaviors of the PCEs, and the results are presented in Figure 12.

As shown in Figure 12, the adsorption of PCEs on cement conforms to a typical Langmuir monolayer adsorption model, namely an increase in the adsorbed amount with PCE dosage along with an adsorption plateau at a saturated adsorption amount. For the w/c = 0.29 fcps, the adsorbed saturated dosage of the PCEs is 0.12%, and the saturated adsorption amount is about 0.47 mg/g. For the w/c = 0.50 fcps, the saturation adsorption amount is 0.37 mg/g at a dosage of 0.12%. The higher saturation adsorption amount of the PCEs at w/c = 0.29 fcps than at w/c = 0.50 fcps is posited to originate from a higher concentration of PCEs in the solution, which may lead to a lower rate of hydration and reduced heat of cement hydration at early ages. Furthermore, the initial fluidity at the lower w/c was more sensitive to the PCE dosage than that at the higher w/c; thus, this result is fully consistent with the WFT variation at different w/c ratios. Additionally, the adsorption amount of the PCE maybe the decisive factor for the WFT.

When the PCE was adsorbed on cement particles, the zeta potential of cement paste was also changed at the same time, as shown in Figure 12. The PCE bearing anionic groups can be adsorbed definitely on the cement paste with positive zeta potential. The curves in Figure 12 behave in a totally inverted way compared with the adsorption curves. Initially, the zeta potentials of both w/c ratios fcps significantly decrease with increasing PCE dosages in the solution. Once beyond the adsorption saturated dosage of the PCE at w/c = 0.29 fcps, the value of zeta potential stops changing, and this phenomenon is fully consistent with the trend of adsorption that no further adsorption of the PCE occurs once the saturated adsorption is achieved. For the w/c = 0.50 fcps, the phenomena are the same as that of 0.29 w/c fcps except that the negative value of zeta potential is larger. This might be attributed to the more adsorption amount of the PCE on the cement particles at the 0.29 w/c fcps.

In order to further study the interaction between the PCE and cement hydration, more specific investigations were performed on the evolution of adsorption behaviors until the setting of the paste, and the results are shown in Figure 13. The results show that the higher w/c leads to the lower adsorption amount. Along with hydration, the adsorption amount of the PCE significantly increases, and it indicated that the un-adsorbed PCE remaining in the liquid phase can further adsorb on the newly formed surface.

### 3.8. Hydration Kinetics Measurement by Calorimetry

Calorimetry was used to study the impacts of the PCEs on the initial hydration of cement. Figure 14a,b shows the initial heat of cement hydration, attributing to the initial dissolution of clinker phases and the formation of initial hydration products [39,40]. The maximum rate of heat generation decreases with increasing PCE concentration, which can be explained by the deceleration of the diffusion of water and ions at the cement–solution interface due to the adsorbed PCE layers. In addition, the water contents of fcps differ significantly between the w/c ratios, and the total heat values of cement hydration within the first 2 h are broadly similar without the PCE. However, a slight difference of heats of cement hydration was observed for pastes at different w/c ratios with the same dosage of PCE. Additionally, a higher w/c results in higher heat release during initial hydration period. Therefore, the inhibition of hydration by the PCEs was more significant at the lower w/c. This may be attributed to the stronger adsorption ability of the PCEs at the lower w/c.

### 3.9. Chemically Bound Water

The development of various products in hydrating cement pastes was quantified by chemically bound water. These results are shown in Figure 15. In Figure 15a,b, it shows that the chemically bound water of the solid phases in the fcps grows with hydration time. Additionally, higher bound water content in lower w/c pastes is observed, which reflects the higher product content in the lower w/c pastes during the early stages. This is partially due to a higher w/c causing a lower hydration ion contents in the cement solution, leading to a lower rate of hydration at early ages. Furthermore, it is clearly seen that the chemically bound water of the solid phases in the w/c = 0.29 fcps stays rather stable in the first two hours, while the hydrating in the w/c = 0.50 fcps rapidly grows with hydration time.

Different evolutions of the chemically bound water were caused by different PCEs. Basically, the fcps containing PCEs show slow growth in chemically bound water, implying retardation effect of the PCEs on cement hydration [32]. At w/c = 0.29, without the PCE, the ratio of bound water content was 6.64% after hydration of 2 h. In the presence of the PCE dosage of 0.2%, the ratio of bound water content decreased to 3.20%. Compared to two ratios of bound water content, it decreased by 51.8%. While at w/c = 0.50, without the PCE, the ratio of bound water content was 5.22% after hydration 2 h. In presence of the PCE with dosage of 0.2%, the ratio of bound water content decreased to 2.68%. Compared to two ratios of bound water content, it decreased by 48.6%. Therefore, with the same amount of the PCEs, a higher degree of hydration in lower w/c pastes is observed; however, the inhibition of hydration in the presence of PCEs in lower w/c pastes is more readily observable than that in higher w/c pastes, which might be attributed to the higher adsorption amount of the PCEs in lower w/c pastes.

### 3.10. XRD and TG Analyses

XRD measurements were conducted to determine the influences of the PCEs on cement hydration, especially on the the AFt content in the cement pastes. According to Figure 16a, the formation of AFt crystals appears to be rare at a hydration time of 5 min but is clearly observed at a hydration time of 2 h, and the AFt contents are comparable between the two pastes at various w/c ratios. Combining XRD (Figure 16a) results with chemically bound water results are shown in Figure 15, it is concluded that the increase in chemically bound water of the fcps can be attributed to the formation of AFt with hydration time and the amounts of AFt between the two w/c ratios are broadly similar.

AFt contents in the presence of PCEs were marked in Figure 16b. Samples in the presence of PCEs all exhibit lower AFt content than that without PCEs, presenting strong inhibition to cement hydration. In addition, the peak intensity of AFt at w/c = 0.29 is slightly higher than that at w/c = 0.50.

The TG analysis supports the mineralogical observations that were obtained via XRD. TG/DTG curves of cement in the absence and presence of PCEs at two w/c ratios are plotted in Figure 17. At 5 min of hydration, the incorporation of PCEs in fcps slightly depresses the amounts of Aft and this result is consistent with the results of calorimetry, chemically bound water and XRD. In addition, the degree of hydration is very low, and the amounts of hydration products in the two w/c pastes are similar. Therefore, the initial fluidity is not controlled by the hydration of cement in the systems with different w/c values. In comparison, at 2 h of hydration, the sample that contains the PCEs at w/c = 0.50 exhibits the smallest mass loss, which is in accordance with the results of chemically bound water.

Since the amount of AFt is defined as a mass loss at temperatures ranging from 50 to 120 °C, and the amount of chemically bound water is defined as a mass loss at temperatures ranging from 50 to 550 °C [24]. The mass losses in these two temperature ranges are listed in Table 5, and the results showed that the amount of AFt and chemically bound water at 5 min was too small and the deviation was large. Therefore, we mainly discuss the mass loss of 2 h. Both AFt and chemically bound water at w/c = 0.29 without adding the PCE were higher than that of the 0.50 water cement ratio. The amount of AFt and chemically bound water with the PCE dosage of 0.20% decreased by 0.10% and 0.21%, respectively, at the w/c = 0.29, but decreased by 0.04% and 0.12%, respectively, at the w/c ratio of 0.50. The TG analysis further supports that the hydration inhibition ability in the presence of PCEs at a lower w/c is stronger than that with a higher w/c mix. Our experiment shows that a lower w/c leads to a larger adsorption amount of the PCEs, stronger inhibition of the hydration of cement and better fluidity retention when compared with the larger w/c system.

## 4. Conclusions

A comparative study of various PCEs dosages and w/c ratios was conducted with respect to their impacts on the rheology, fluidity and initial hydration process of fcps, aiming at correlating the dispersing capability with the WFT, zeta potential and hydration products in cement pastes. The results indicate that:The rheological parameters decreased, and the flow parameters increased while the PCE dosages and WFT increased. However, the effects of the PCE on the rheological and flow properties of fcps differed for various w/c values. Especially for a lower w/c, the incorporation of the PCE had huge influence on rheology, flow parameters, WFT and zeta potential. This might be attributed to the more adsorption amount of the PCE on cement particles at a lower w/c than that at a higher w/c. Furthermore, the adsorption amount of the PCE mainly affects the workability of the paste and a certain increase in water exhibits a negligible influence on it.From the results of calorimetry, chemically bound water, XRD and TG analysis, the early hydration degree of fcps at different w/c ratios is broadly similar and the cement paste has a slightly higher hydration rate at a lower w/c than at a higher w/c. This can be attributed to the higher ion concentration at a lower w/c than that at a higher w/c.The early formation of the AFt crystals was largely changed by the PCE and this is the most important factor in the fluidity retention of fcps. Chemically bound water, XRD and TG measurements show that during the first 2 h the incorporation of the PCE significantly slows down the early formation of Aft crystals, particularly in a lower w/c paste due to a larger adsorption amount of the PCE at a lower w/c. Furthermore, at the same dosage of the PCEs, the cement paste has a higher hydration rate at a lower w/c than at a higher w/c, although larger adsorption amounts of the PCEs at a lower w/c correspond to stronger inhibition of hydration. Therefore, if the PCE dosage is the same at various w/c ratios, the fluidity loss of the cement paste is much more dramatic at a higher w/c than at a lower w/c.

## Figures and Tables

**Figure 1 materials-14-03195-f001:**
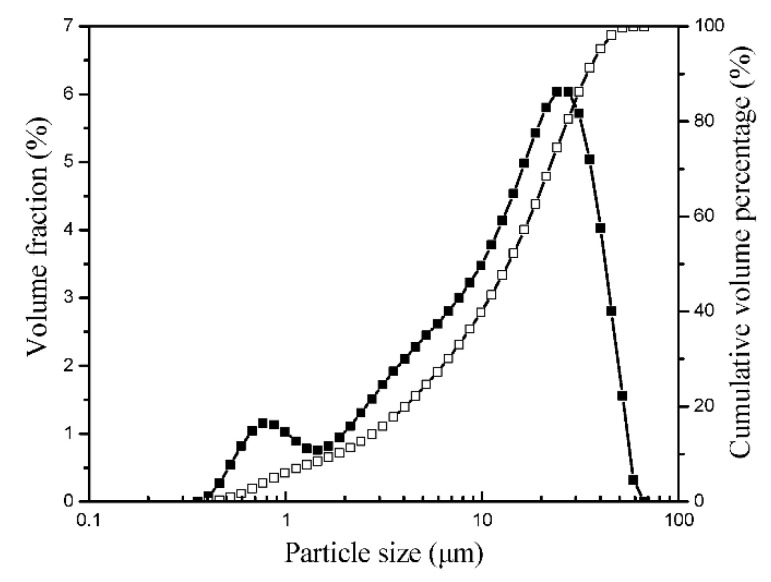
Particle size distribution of the cement.

**Figure 2 materials-14-03195-f002:**
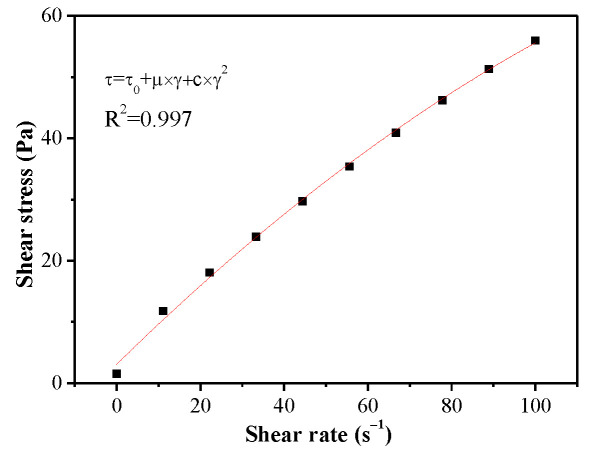
Rheological curves of cement paste samples.

**Figure 3 materials-14-03195-f003:**
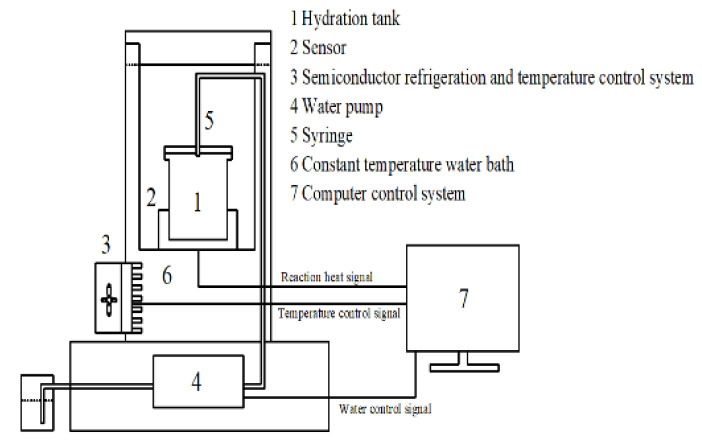
Automatic cement hydration heat tester.

**Figure 4 materials-14-03195-f004:**
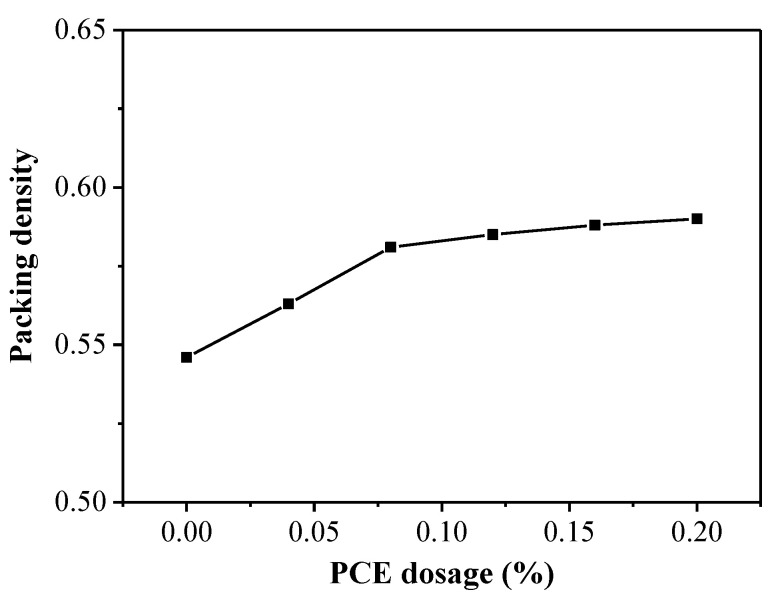
Packing density versus the PCE dosage for fresh cement paste.

**Figure 5 materials-14-03195-f005:**
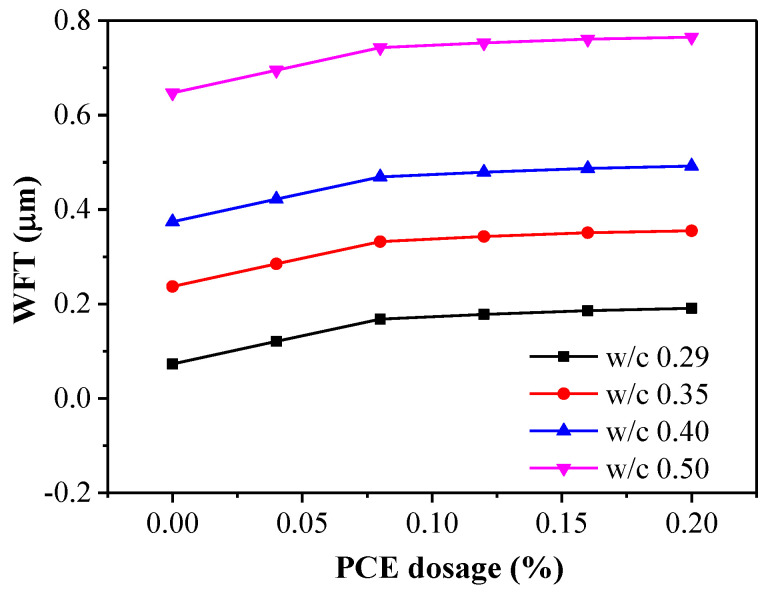
WFT versus PCE dosage for pastes with various w/c ratios.

**Figure 6 materials-14-03195-f006:**
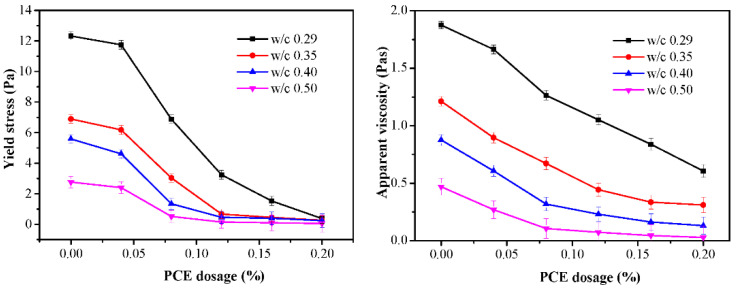
Variations of the yield stress and apparent viscosity with w/c for different PCE contents.

**Figure 7 materials-14-03195-f007:**
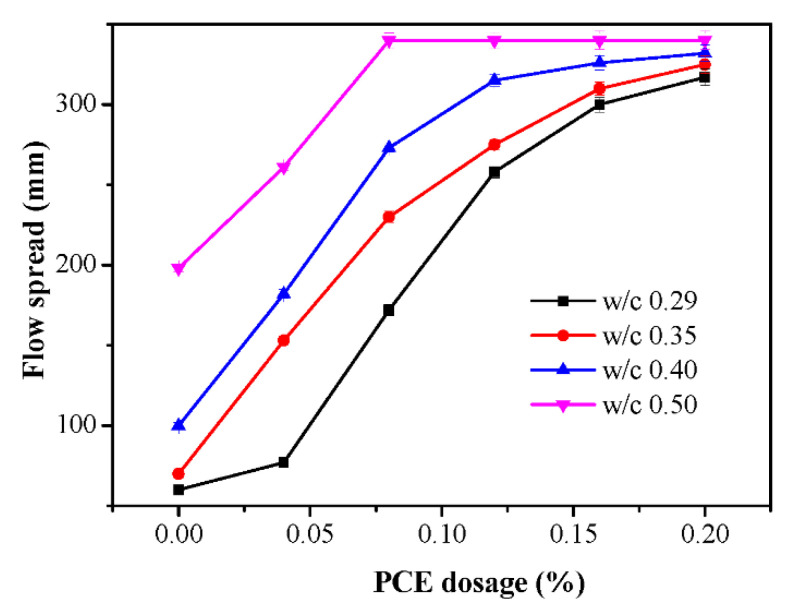
Variations of the flow spread with w/c for different PCE contents.

**Figure 8 materials-14-03195-f008:**
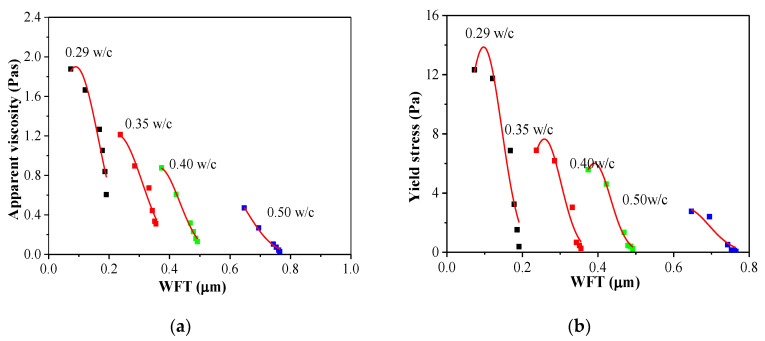
Yield stress and apparent viscosity versus WFT. (**a**) Yield stress versus WFT; (**b**) Apparent viscosity versus WFT.

**Figure 9 materials-14-03195-f009:**
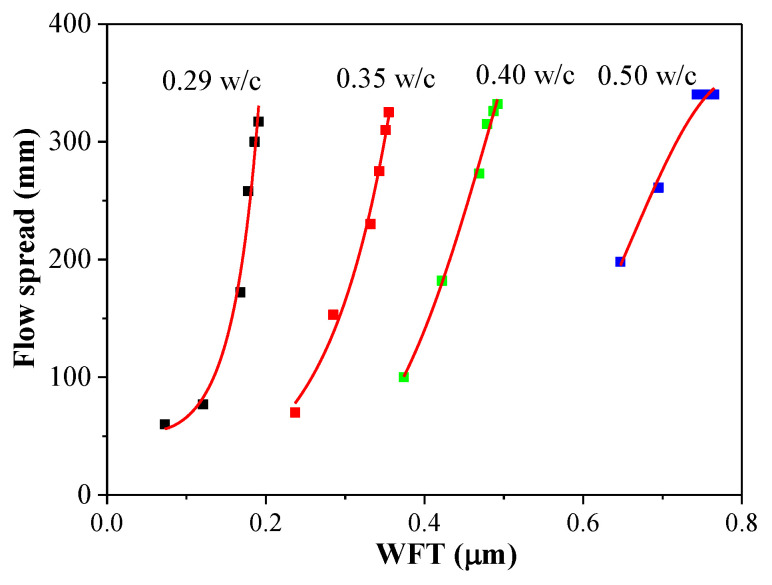
Flow spread versus WFT.

**Figure 10 materials-14-03195-f010:**
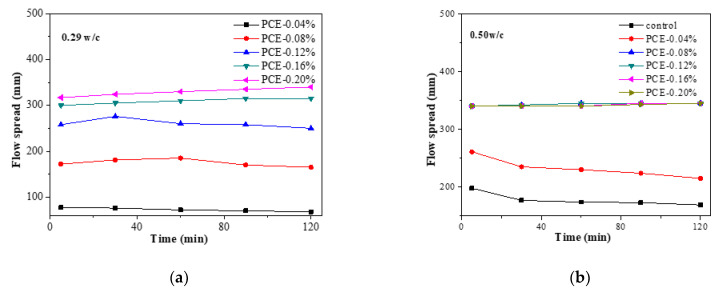
Influences of PCEs on the fluidity retention of cement pastes at two w/c ratios: (**a**) w/c = 0.29 and (**b**) w/c = 0.50.

**Figure 11 materials-14-03195-f011:**
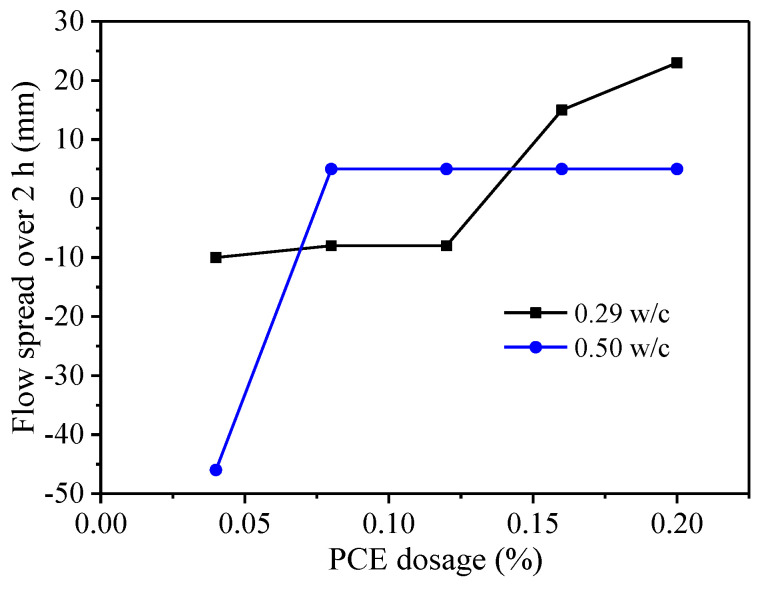
Time change of flow spread over 2 h.

**Figure 12 materials-14-03195-f012:**
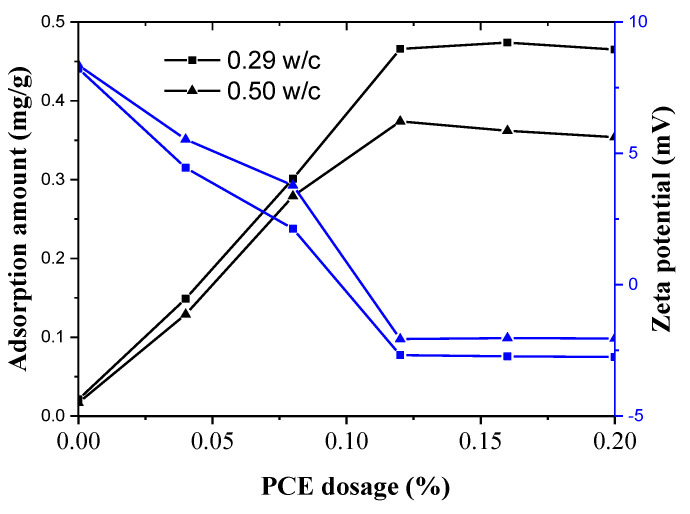
Adsorption behaviors and zeta potentials of the PCEs in the two w/c cement pastes.

**Figure 13 materials-14-03195-f013:**
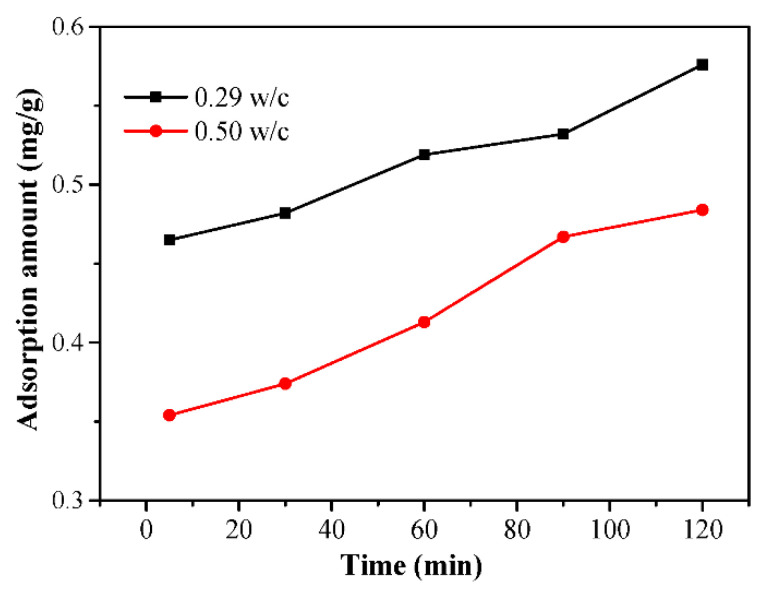
Adsorption amount evolution of the PCEs in the two w/c cement pastes over the elapsed time.

**Figure 14 materials-14-03195-f014:**
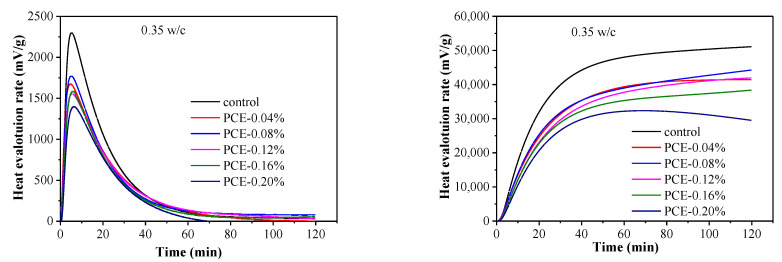
Calorimetry curves of cement pastes in the presence of PCEs at various dosages at 25 °C: (**a**) differential heat flow and (**b**) cumulative heat flow.

**Figure 15 materials-14-03195-f015:**
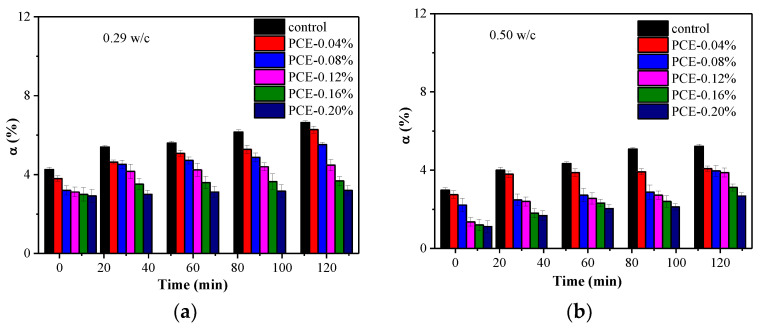
Bound water of samples at two w/c ratios (normalized to ignited mass): (**a**) w/c = 0.29 and (**b**) w/c = 0.50.

**Figure 16 materials-14-03195-f016:**
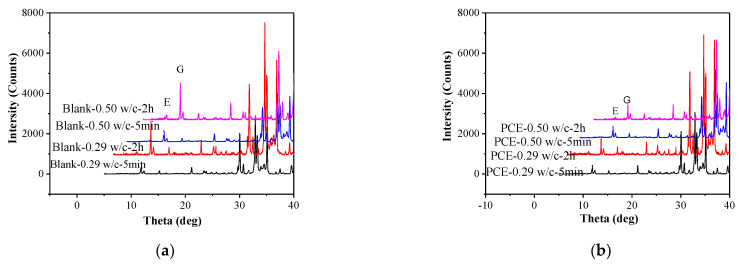
XRD spectra of the cement pastes at various w/c ratios with (**a**) and without PCE (**b**).

**Figure 17 materials-14-03195-f017:**
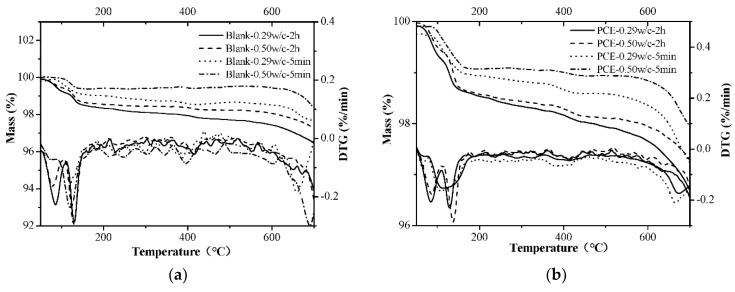
TG analysis of the cement pastes at two w/c ratios without (**a**) and with PCEs (**b**).

**Table 1 materials-14-03195-t001:** Chemical and mineral compositions of the cement.

Chemical Composition (Mass %)									Mineral Composition (Mass %)			
SiO_2_	Al_2_O_3_	Fe_2_O_3_	CaO	MgO	SO_3_	Na_2_O_eq_	f-CaO	Cl^−^	C_3_S	C_2_S	C_3_A	C_4_AF
22.01	4.47	3.45	64.31	2.45	2.45	0.512	0.90	0.01	56.26	19.99	6.13	11.00

**Table 2 materials-14-03195-t002:** Packing density and WFT results of paste mixes.

Mix No.	Packing Density	Water Ratio	Excess Water Ratio	Specific Surface Area (m^2^/m^3^)	Water Film Thickness (μm)
C–0.00–0.29	0.546	0.916	0.084	1,156,000	0.073
C–0.04–0.29	0.563	0.916	0.146	1,156,000	0.121
C–0.08–0.29	0.581	0.916	0.194	1,156,000	0.168
C–0.12–0.29	0.585	0.916	0.206	1,156,000	0.178
C–0.16–0.29	0.588	0.916	0.240	1,156,000	0.186
C–0.20–0.29	0.590	0.916	0.249	1,156,000	0.191
C–0.00–0.35	0.546	1.106	0.274	1,156,000	0.237
C–0.04–0.35	0.563	1.106	0.336	1,156,000	0.285
C–0.08–0.35	0.581	1.106	0.384	1,156,000	0.332
C–0.12–0.35	0.585	1.106	0.396	1,156,000	0.343
C–0.16–0.35	0.588	1.106	0.431	1,156,000	0.351
C–0.20–0.35	0.590	1.106	0.439	1,156,000	0.355
C–0.00–0.40	0.546	1.264	0.432	1,156,000	0.374
C–0.04–0.40	0.563	1.264	0.494	1,156,000	0.422
C–0.08–0.40	0.581	1.264	0.543	1,156,000	0.469
C–0.12–0.40	0.585	1.264	0.555	1,156,000	0.479
C–0.16–0.40	0.588	1.264	0.589	1,156,000	0.487
C–0.20–0.40	0.590	1.264	0.597	1,156,000	0.492
C–0.00–0.50	0.546	1.580	0.748	1,156,000	0.647
C–0.04–0.50	0.563	1.580	0.990	1,156,000	0.695
C–0.08–0.50	0.581	1.580	1.075	1,156,000	0.743
C–0.12–0.50	0.585	1.580	1.116	1,156,000	0.753
C–0.16–0.50	0.588	1.580	1.124	1,156,000	0.761
C–0.20–0.50	0.590	1.580	1.130	1,156,000	0.765

**Table 3 materials-14-03195-t003:** The fitting equations and R^2^ value of yield stress and apparent viscosity versus WFT.

w/c Ratios	Yield Stress Versus WFT		Viscosity Versus WFT	
	Fitting Equations	R^2^	Fitting Equations	R^2^
0.29	τ = e^(0.55 + 42.68^ ^× WFT − 219.32^ ^×^ ^(WFT)2)^	0.888	τ = e^(−0.05 + 15.44^ ^×^ ^WFT − 86.22^ ^×^ ^(WFT)2)^	0.905
0.35	τ = e^(−14.79 + 130.45^ ^×^ ^WFT − 253.84^ ^×^ ^(WFT)2)^	0.926	τ = e^(−3.55 + 33.132^^×^ ^WFT − 73.16^ ^×^ ^(WFT)2)^	0.931
0.40	τ = e^(−39.72 + 212.73^ ^×^ ^WFT − 272.49^ ^×^ ^(WFT)2)^	0.991	τ = e^(−13.87 + 75.64^ ^×^ ^WFT −104.01^ ^×^ ^(WFT)2)^	0.988
0.50	τ = e^(−48.58 + 157.54^ ^×^ ^WFT − 124.91^ ^×^ ^(WFT)2)^	0.860	τ = e^(−24.86 + 84.3^ ^×^ ^WFT − 72.73^ ^×^ ^(WFT)2)^	0.984

**Table 4 materials-14-03195-t004:** The fitting equations and R^2^ value of flow spread versus WFT.

w/c Ratios	Flow Spread Versus WFT	
	Fitting Equations	R^2^
0.29	τ = e^(−10.87 + 41.97^ ^×^ ^WFT − 26.30^ ^×^ ^(WFT)2)^	0.976
0.35	τ = e^(−4.25 + 33.94^ ^×^ ^WFT − 27.39^ ^×^ ^(WFT)2)^	0.992
0.40	τ = e^(1.71 + 10.63^ ^×^ ^WFT + 2.34^ ^×^ ^(WFT)2)^	0.983
0.50	τ = e^(4.30 − 11.09^ ^×^ ^WFT + 98.99^ ^×^ ^(WFT)2)^	0.927

**Table 5 materials-14-03195-t005:** Weight loss (%) of the pastes in the range 50–120 °C and 50–550 °C.

Sample	5 min		2 h	
	50–120 °C	50–550 °C	50–120 °C	50–550 °C
Blank-0.29 w/c	0.41	1.16	0.86	2.35
Blank-0.50 w/c	0.31	0.97	0.66	1.76
PCE0.2%-0.29 w/c	0.39	1.03	0.76	2.12
PCE0.2%-0.50 w/c	0.28	0.92	0.62	1.65

## Data Availability

Some or all data, models or code that support the findings of this study are available from the corresponding author upon reasonable request.

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
