# Peer review of "Effects of PCE on the Dispersion of Cement Particles and Initial Hydration"

_materials, 2021, doi:10.3390/ma14123195_

Round 1

Reviewer 1 Report

It was a pleasure to read and review this article. The goal of this manuscript is to experimentally investigate the effects of PCEs dosages and w/c ratios in terms of rheology, fluidity and initial hydration process of fcps, aiming at correlating the dispersing capability with the WFT, zeta potential and hydration products in cement pastes. The research objectives and methods are sound. The following minor comments might help to improve the quality of the manuscript:

  • Introduction: The authors refer to too many redundant references [8-15, 17-21] without critical reviews.
  • Section 3.1: Just 8 % increase is not significant.
  • Section 3.2: It is difficult to agree with the authors’ claim the ‘a strong increase in WFT occurs in the case of the lower w/c’ because the difference between the comparison is the same as 0.118 (=0.191-0.073) and 0.118 (= 0.765-0.647).
  • Section 3.5.1: It is not clear why the authors adopt a certain type of exponential curve for the regression analysis rather than linear lines.
  • Section 3.6: The second part of the section is not a fair comparison because of the different dosages of PCEs.
  • Section 3.8: The strength development is also needed to be presented for better understanding.
  • The last paragraph of Section 3.10: The testing methods for chemically bound water and TG analysis are based on the same testing technique. Therefore, the comparison based on the same testing technique is not meaningful.

Reviewer 2 Report

This paper shows the effects of PCE on the dispersion of cement particles and initial hydration. My major concern is that this type of work has already been done by many other researchers. Using TG and XRD techniques are not novel for assessing hydration of cementitious material. The authors need to re-claim the novelty of this work. 

Another concern is that the materials studied lacks of generality. This work is based on specific type of cement and PCE. What if other type of GP and /or PCE are used ? 

TG and XRD were done well. But the analyses of results need bit more work. For instance, TG curve is ok for the readers. Can the authors also calculate the degree of hydration based on chemically bonded water and also calcium hydroxide content in each mix

I would like to suggest a major round of revision for the authors to consider the above comments

Reviewer 3 Report

Comb-like polycarboxylate (PC) superplasticizers are recently developed 31 and have become an integral part of modern concrete technology [6, 7] - Comb-like polycarboxylate (PCE) superplasticizers are recently developed 31 and have become an integral part of modern concrete technology.

Isothermal calorimeter – interesting.

Figure 4. Packing density versus the PCE dosage for fresh cement paste – whether it's average results?

The results of the yield stress and apparent viscosity are plotted against the PCE dos- 238 ages for different w/c ratios – whether they confirm the research carried out by other authors.

Figure 8. Yield stress and apparent viscosity versus WFT – The drawing is indistinct. Perhaps it would be better to present it in the form of a table and figure?

Influences of PCEs on the fluidity retention of cement pastes at two w/c ratios: (a) w/c 335 =0.29 and (b) w/c =0.50. - what is happening with different w/c ratios - please describe. What was the guiding principle behind the selection of only two presented ration. What about the rest 2 rations?

3.7. Adsorption behavior of the PCEs and the dispersing power per adsorption amount on cement – what about it is seed by other literature

Figure 14. Calorimetry curves of cement pastes in the presence of PCEs at various dosages at 25 434 °C: (a) differential heat flow and (b) cumulative heat flow. – for two w/c=0.35 I 0.5 what about the rest?

Round 2

Reviewer 2 Report

The authors have properly addressed my comments. The paper is now recommended as a publication in the journal.

Reviewer 3 Report

Thank you for considering your comments